# Development of a histopathology scoring system for the pulmonary complications of organophosphorus insecticide poisoning in a pig model

Elspeth J. Hulse[1,2], Sionagh H. Smith[3]*, William A. Wallace[4,5], David A. Dorward[5], A. John Simpson[2,4], Gordon Drummond[6], Richard E. Clutton[7], Michael Eddleston[1]

1 Pharmacology, Toxicology & Therapeutics, Centre for Cardiovascular Science, University of Edinburgh, Edinburgh, United Kingdom, 2 Translational and Clinical Research Institute, Newcastle University, Royal (Dick) School of Veterinary Medicine, University of Edinburgh, Edinburgh, United Kingdom, 3 Easter Bush Pathology, Royal (Dick) School of Veterinary Medicine, University of Edinburgh, Edinburgh, United Kingdom, 4 Centre for Inflammation Research, University of Edinburgh, Edinburgh, United Kingdom, 5 Department of Pathology, University of Edinburgh, Edinburgh, United Kingdom, 6 Anaesthesia, Critical Care and Pain Medicine, Division of Health Sciences, University of Edinburgh, Edinburgh, United Kingdom, 7 Wellcome Critical Care Laboratory for Large Animals, Royal (Dick) School of Veterinary Medicine, University of Edinburgh, Edinburgh, United Kingdom

* Sionagh.Smith@ed.ac.uk

**Data Availability Statement:** All relevant data are within the paper and its Supporting Information files.

## Abstract

Organophosphorus (OP) insecticide self-poisoning causes over 100,000 global deaths annually. Around a third of patients are intubated and up to half of these can die. Post-mortem analysis of OP poisoned patients' lungs reveals consolidation, edema and hemorrhage, suggesting that direct or indirect lung damage may contribute to mortality. The lung injury caused by these formulated agricultural preparations is poorly characterised in humans, and a valid histopathology scoring system is needed in a relevant animal model to further investigate the disease and potential treatments. We conducted two pilot studies in anesthetized minipigs, which are commonly used for toxicological studies. In the first, pigs were given 2.5 mL/kg of either OP (n = 4) or saline (n = 2) by gavage and compared with positive controls (iv oleic acid n = 2). The second study simulated ingestion followed by gastric content aspiration: mixtures of OP (n = 3) or saline (n = 2) (0.63–0.71mL/kg) were placed in the stomach, and then small volumes of the gastric content were placed in the lung. At post-mortem examination, lungs were removed and inflation-fixed with 10% neutral buffered formalin. Samples (n = 62) were taken from cranial and caudal regions of both lungs. Two experienced lung histopathologists separately scored these samples using 8 proposed features of damage and their scores related (Kendall rank order). Two elements had small and inconsistent scores. When these were removed, the correlation increased from 0.74 to 0.78. Eight months later, a subset of samples (n = 35) was re-scored using the modified system by one of the previous histopathologists, with a correlation of 0.88. We have developed a reproducible pulmonary histopathology scoring system for OP poisoning in pigs which will assist future toxicological research and improve understanding and treatment of human OP poisoning.

**Funding:** The studies were funded by grants 085979, 090886 and 104972 from the Wellcome Trust, as well as by a Lister Research Prize Fellowship and Scottish Senior Clinical Fellowship awarded to Professor Michael Eddleston, University of Edinburgh.

**Competing interests:** The authors have declared that no competing interests exist.

## Introduction

Pesticides have long been used for public health purposes and to protect global food production. The US spends $15 billion a year on pesticides [1], the most common of which is the herbicide glyphosate [2]. Sadly, ingestion of pesticides remains one of the most common methods for suicide, causing over 100,000 deaths per year globally, particularly in rural Asia [3]. Two thirds of deaths follow self-poisoning with OP compounds [4], which primarily act by inhibiting the enzyme acetylcholinesterase (AChE) in the central and autonomic nervous systems, neuromuscular junctions, red cell membranes and lungs [5,6]. AChE inhibition increases cholinergic activity in both central and peripheral nervous systems. Acute features of poisoning include reduced consciousness, bronchorrhea, bronchospasm, diarrhoea, vomiting and paralysis [7]. Death is usually from respiratory failure [8,9]; its incidence depends on the toxicity and chemistry of the compounds ingested, their pharmacokinetics, and the availability of sufficient healthcare to both prevent pre-hospital death and treat complications of poisoning [10]. Approximately one third of poisoned patients admitted to hospital require intubation and ventilation due to their symptoms, and mortality of intubated patients is high, ranging from 23–50% [11–13].

Many patients lose consciousness before reaching hospital, and may aspirate regurgitated stomach contents, leading to severe lung injury [14,15]. It is also possible that systemically absorbed OP damages the lung by disrupting the integrity of the alveolar capillaries. In a large Indian autopsy study of OP poisoned patients (n = 85), 75% (27/36) of patients who died within 24 hours had pulmonary interstitial edema and 25% had parenchymal hemorrhage. Of those dying after 24 hours, 69% (34/49) had lobar or segmental consolidation and 30% (15/49) had interstitial edema [16]. [NB; it is possible a small number of these autopsies had carbamate poisoning instead of true organophosphate poisoning and additional toxins such as alcohol and kerosene were consumed]. A comprehensive review of respiratory complications secondary to OP compound poisoning is available elsewhere [15].

To investigate the effects of both aspirated and systemically absorbed OP on the lung, we developed a large animal model to allow measurement of both direct and indirect lung injury. We chose the Gottingen minipig because it is used extensively in toxicological research [17], its lungs are similar to human lungs [18], and bronchoscopy, lavage, and pulmonary biopsy are practical. Rodents have plasma carboxylesterase (unlike pigs and humans) which could theoretically allow detoxification of OP compounds more efficiently and so were not used in this study [19,20].

We searched for a validated pulmonary histopathological scoring system for OP poisoning in the pig but found only unvalidated systems developed for small animals such as rats and rabbits [21,22]. Although pulmonary histopathology scoring systems exist for gastric aspiration injury, none have been used to assess changes after aspiration of OP compounds and gastric juices. The aim of this study was to design a reproducible scoring system to measure both indirect (systemic) and direct (via placement of OP insecticide and gastric juice in lungs) histological lung damage using material from two separate pilot studies conducted by our research team. The new pulmonary histopathological scoring system will assist further porcine aspiration studies and could inform observational human OP insecticide poisoning work in the future.

## Materials and methods

Pulmonary histopathology samples for this work were obtained from two large animal pilot studies (Eddleston, in preparation) investigating indirect lung injury through systemic poisoning by gavage (study 1) and simultaneous indirect and direct lung injury through poisoning by

gavage and pulmonary placement (study 2) using a common WHO class II moderately toxic OP insecticide, dimethoate.

The tissue samples were then scored by two independent histopathologists using a specifically designed scoring system. The reliability of the scoring system was assessed by demonstrating repeatability through inter and intra observer statistical correlation.

## Animal handling

The studies were performed under an approved Home Office Licence (PPL60 3757) after institutional ethics review (Moredun Research Institute, Pentlands Science Park, Midlothian). All experiments used adult male Gottingen minipigs (Ellegaard Minipigs ApS, Dalmose, Denmark). Animals were barrier bred and confirmed free of infections before shipment to the study site. The pigs were housed in pens with liberal access to food and water under the care of veterinary surgeons. Food was withheld for one night before a study. The animals were treated in accordance with the Animals (Scientific Procedures) Act of 1986.

## Anesthesia, instrumentation and monitoring

In both studies anesthesia, instrumentation and monitoring were performed as previously described [23] with some differences detailed below.

In study 1, pigs were ventilated with $F_IO_2$ >0.95 whereas study 2 used a 50:50 oxygen: air ratio. Both carrier gas combinations were mixed with the volatile agent isoflurane. Study 1 used a Manley minute volume divider ventilator. In study 2, the pigs' lungs were protectively ventilated using tidal volumes <8 ml/kg delivered by a Servo 300 ICU ventilator (Maquet, Sweden). In study 2, minipigs were transported to a CT scanner at regular intervals to allow lung volume and radiological density to be measured. The studies ended when the anaesthetized pigs were killed with intravenous pentobarbital (40 mg/kg).

**Study 1 design and interventions.** This 12 hr study was conducted in eight male minipigs (mean weight 21.4 [SD 2.1] kg) that received 2.5 mL/kg of a 40% emulsifiable concentrate formulation of dimethoate (dimethoate EC40, BASF SE, Ludwigshafen, Germany; n = 4) or saline (n = 2) by gavage with two pigs receiving a dose of intravenous oleic acid as positive controls (0.25 mL/kg in the first pig and 0.17 mL/kg in the second pig–the second dose was reduced due to the severe toxicity in the first pig). The pigs given insecticide were also given IV oxime therapy (pralidoxime loading dose 20 mg/kg followed by an infusion 15.5 mg/kg/hr) to reactivate AChE, similar to treatment of human OP poisoning. All minipigs received a single-sided bronchoalveolar lavage (BAL) at baseline (time -30 min) and at 4 and 12 hr using a 60 mL sterile saline aliquot introduced through the working channel of a bronchoscope (VETVU VFS-2A Veterinary fiberscope, Krusse, UK). Bronchial biopsies were also taken at 12 hr.

**Study 2 design and intervention.** This study lasted 48 hr and was conducted in five male minipigs (mean weight 20.6 [SD 0.96] kg) that received 1/5[th] of the previously noted toxic oral dose of dimethoate EC40 (to cause less cardiotoxicity and allow pulmonary injury to develop over 48 hr).

Three minipigs were given 0.6–0.7 mL/kg dimethoate EC40 through an orogastric tube and, after 30 min, the gastric contents (GC) were aspirated using the orogastric tube and placed into a lung using the working channel of the bronchoscope. Two further minipigs were given a similar volume of 0.9% sodium chloride by orogastric administration, with their gastric contents placed into a lung 30 min later.

The first pig was given 50 mL of a mixture of OP insecticide and gastric contents into the main bronchus of one lung. This caused cardiovascular collapse and euthanasia was required.

**Table 1. Interventions used for pigs in Study 2.**

| Pig | Intervention | Dose given by gavage | Volume of gastric contents administered to lung |
| --- | --- | --- | --- |
| 1 | OP + GC | dimethoate EC40 0.65 mL/kg (13 mL) | 50 mL |
| 2 | Saline + GC | saline 0.63 mL/kg (15 mL) | 10 mL |
| 3 | OP + GC | dimethoate EC40 0.63 mL/kg (13 mL) | 10 mL |
| 4 | OP + GC | dimethoate EC40 0.71 mL/kg (15 mL) | 10 mL |
| 5 | Saline + GC | Saline 0.68 mL/kg (15 mL) | 10 mL |

**Abbreviations:** OP; organophosphorus insecticide, GC; gastric contents, EC; emulsifiable concentrate.

The dose was therefore modified for the remaining animals (Table 1) such that they received 10 mL of gastric contents administered into a lung.

## Histopathology

At post-mortem examination, the lungs were removed and inflation-fixed via the trachea with 10% neutral buffered formalin (NBF) to a fluid pressure of approximately 25 cm $H_2O$. The tracheas were ligated and lungs immersed in NBF. Lung tissue was sampled as follows: in the first study, both cranial and caudal lobes and the right middle lobe were sampled (the right caudal lobe was inadvertently not sampled in one oleic acid pig); in the second study, both caudal lobes, the left cranial and, where possible, the right middle lobes were sampled, producing a total of four or five samples per pig. In order to better typify lung injuries, two samples were collected from an obvious area of lung injury in four pigs. The samples were stored in 10% NBF prior to processing and paraffin wax embedding. Four-micron sections were stained with hematoxylin and eosin (HE). Lung sections (approximately 2-4cm$^2$) containing bronchi and bronchioles were placed on slides and dried for 15 min at 37˚C, then 60˚C for 25 min.

## Rationale for development of histopathology scoring system

We first reviewed the literature for validated scoring systems for OP poisoning. Pulmonary histopathology from animal studies showed that exposing lungs to OP insecticide (directly or indirectly) causes edema, hemorrhage, alveolar destruction and inflammation [21,22,24–26]. Two of the three scoring systems described in these studies used a 4-point semi-quantitative system [21,26] while other studies did not score the histological changes at all, choosing to take a qualitative, descriptive approach instead [24,25]. Only one of the three studies that used a pulmonary histopathology scoring system [21,22,26] reported the location and number of samples taken [21].

Yavuz et al used a detailed pulmonary histopathology scoring system [22] to study the effects of orogastric OP administration in rabbits. It evaluated eight histopathological features, each feature being awarded a score from 0 to 5. We did not use this scoring method because it was based on rabbit studies that used partial liquid ventilation [27] and on rats with lithium induced lung toxicity [28]. Thus, for most published studies, neither the species of animal nor the mechanism of injury was appropriate for our needs. We also wished to include features of aspiration lung injury in our scoring system [29].

**Development of the pulmonary histopathological scoring system.** Based on our experience of previous assessment of experimental lung injury in animals, we developed a pulmonary histology scoring system to incorporate the cardinal features that we expected to find [19,20,24–26,30]. One author (SS), a board certified veterinary pathologist with multispecies experience in pulmonary histopathology, led the development of the scoring system.

The original histopathology scoring system was modified following initial review (see Results) but it focussed on airway inflammation and incorporated eight indicators of lung injury outlined in Table 2. Each indicator was awarded a score from 0–3 based on different semi-quantitative measures depending on the indicator. This ordinal method of scoring samples allowed a maximum of 24 points per sample, while the modified version allowed up to 18 points. The scores were not based on fields of view analysed, but on the worst score attainable within the whole tissue sample present on the slide, e.g. if there were three bronchioles but only one bronchiole contained neutrophils, the score would be based solely on the bronchiole containing the neutrophils (i.e. not as a mean score of the three bronchioles).

**Inter-observer correlation.** The above-mentioned pathologist (SS) and a human pathology consultant specializing in lung disease (WW) used the original scoring system described

**Table 2. Histopathological scoring system (also see Supporting information).**

| Structure | Lesion | Extent | Score |
|---|---|---|---|
| 1. Bronchial lumens (S1A–S1D Fig) | Neutrophils | None | 0 |
| | | <10 per airway | 1 |
| | | 11–50 per airway | 2 |
| | | >50 per airway | 3 |
| 2. Bronchiolar lumens (S2A–S2D Fig) | Neutrophils | None | 0 |
| | | <10 per airway | 1 |
| | | 11–50 per airway | 2 |
| | | >50 per airway | 3 |
| 3. Bronchial/bronchiolar | Epithelial necrosis /degeneration | None | 0 |
| | | Mild | 1 |
| | | Moderate | 2 |
| | | Severe | 3 |
| 4. Perivascular | Inflammation / fibrin | None | 0 |
| | | Mild | 1 |
| | | Moderate | 2 |
| | | Severe | 3 |
| 5. Alveoli / interstitium (S3A–S3D Fig) | Edema | None | 0 |
| | | <25% | 1 |
| | | 25–50% | 2 |
| | | >50% | 3 |
| 6. Alveoli (S4A–S4D Fig) | Inflammatory cells | None—few | 0 |
| | | Mild increase | 1 |
| | | Moderate | 2 |
| | | Marked | 3 |
| 7. Interstitial (S5A–S5D Fig) | Inflammatory cells | None—few | 0 |
| | | Mild increase | 1 |
| | | Moderate | 2 |
| | | Marked | 3 |
| 8. Anywhere (S6A–S6D Fig) | Hemorrhage / necrosis / fibrin | None | 0 |
| | | Up to 5% of section | 1 |
| | | 5–50% of section | 2 |
| | | >50% of section | 3 |

The above scoring system also allowed for qualitative description of additional features, such as thrombosis, emphysema, fibrosis and vasculitis. Original max score per sample 8 x 3 = 24, modified max score 6 x 3 = 18. The grey sections highlight the criteria removed from the original scoring system to improve inter-scorer correlation.

above to score each lung section (n = 62) using light microscopy. Both independently evaluated the same set of slides in a random order while blinded to the study groups. The initial analysis found that removing two of the low scoring components (see Results) improved the inter-scorer correlation. This created the 'modified' scoring system (Table 2). A subset of lung sections was subsequently re-scored by one histopathologist, using the modified scoring system to calculate the intra-scorer correlation.

**Intra-observer correlation.** A subset of slides was re-scored by one histopathologist to determine repeatability. Briefly, SS blindly re-scored a random selection of samples (n = 35) from studies 1 and 2 using the modified scoring system approximately eight months after the first scoring session.

**Photography.** Images were captured using a BX41 (Olympus, USA) microscope and DP72 camera (Olympus, USA). Image software is Cell^D (Olympus, USA).

**Statistical analysis.** Although utilising the clinical information from two previous expensive large animal pilot studies meant accepting low numbers of animals, it is in keeping with the reduction and refinement principle of 3R's in research [30].

The correlation between scorers was tested using Kendall rank order correlation [31]. This test yields a coefficient T that indicates the difference between the probability that the scores are in the same order, and the probability that the scores have a different order: a direct indication of the weakness of the null hypothesis, and an indication of the strength of association of the scores from the two raters. The significance of T was calculated through z and if greater than 4 would mean p<0.00003, indicating a very strong relationship between raters' scores. The inter and intra-scorer correlations indicated the repeatability of the scoring system [32].

The scores were also used to generate a Bland-Altman pattern of plot [33] using Graph Pad 6.07 (CA, USA) to identify which elements of the original scoring system required removal to improve correlation. Since the data were neither continuous, nor normally distributed, we limited the presentation to show the quartile values for the distributions of the differences, which indicate the magnitude of the inter-observer differences.

## Results

Study 1 showed that OP poisoning by gavage indirectly created lung injury characterized by an increased presence of hemorrhage (within alveoli and within airways) compared to saline controls. Increased numbers of neutrophils were also noted within different compartments of the lung (airways, alveolar spaces, interstitium) in treated pigs compared to controls, as was alveolar and interstitial edema. However, fibrin deposition was found in both control and OP poisoned minipigs. In study 2, placement of gastric contents into the lung, either after OP or saline gavage, created more severe lung injury than the first study.

Fig 1A and 1B illustrate histological features in the lung from a saline pig from study 1 and an OP poisoned pig from study 2, respectively. Representative sections of the modified scoring system are illustrated as Supplementary data.

### Inter-observer correlation

The scores produced by the two scorers showed a strong inter-scorer correlation with a Kendall rank order coefficient T = 0.74, p<0.00003 (Fig 2A). The data had an IQ range of 2 (Fig 2B).

The 'bronchial epithelial necrosis' and 'perivascular inflammation' components of the scoring system were removed as they had low scores across all slides. This created the modified scoring system which resulted in a small improvement in correlation (T = 0.78, p<0.00003) and similar IQ range of 2 (Fig 3A and 3B).

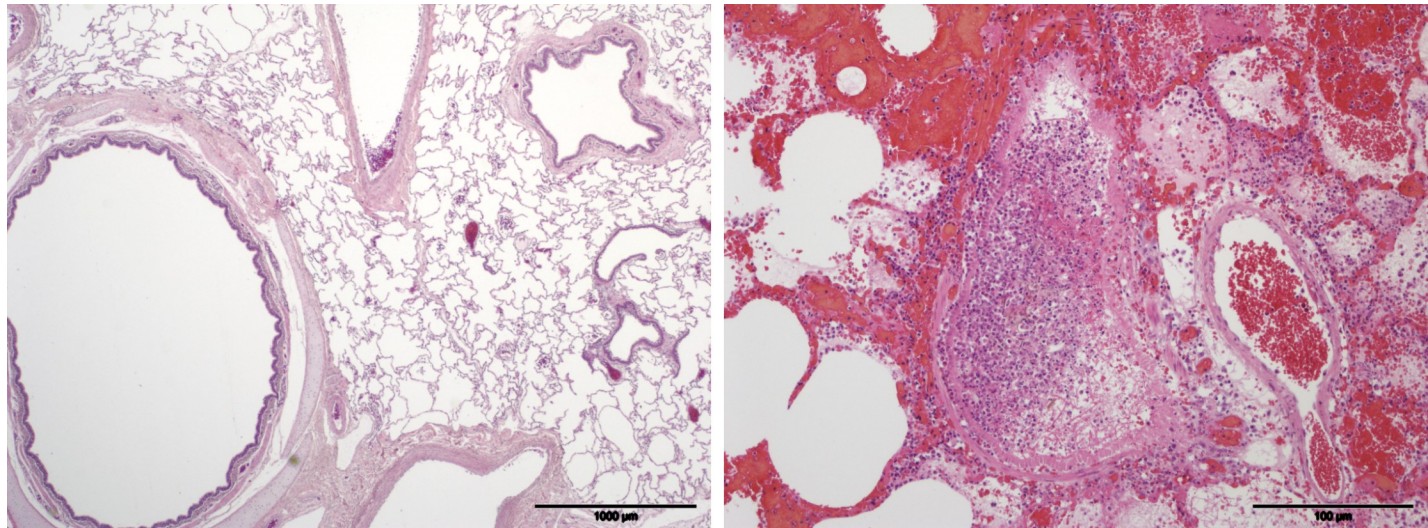

**Fig 1.** Example pulmonary histopathology slides from a study 1 pig given saline by gavage (A) and a study 2 pig given organophosphorus insecticide by gavage then placed in a lung (B). The lung parenchyma and airways in Fig A are within normal limits. In Fig B most of the alveolar spaces are filled with blood, edema fluid and moderate to large numbers of neutrophils. The structure in the centre is a bronchiole filled with neutrophils admixed with fibrin. The bronchiolar epithelial lining is virtually completely necrotic and sloughed. Porcine lung, HE stained.

**Intra-observer correlation.** Fig 4 summarises the intra-observer correlation for one scorer (SS) who blindly re-scored a random selection of samples (n = 35) from both studies eight months after the first scoring session. The correlation (T = 0.88, p<0.00003) was greater than those of comparisons with WW when using the modified scoring system (Fig 3).

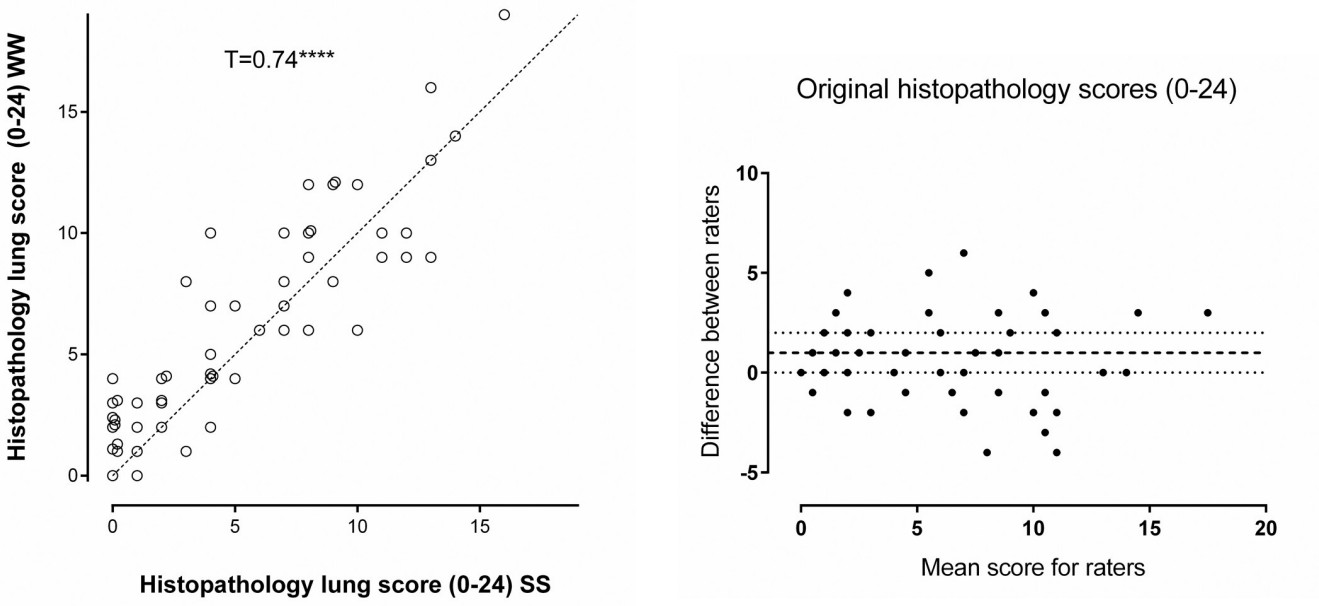

**Fig 2.** A. Scatter dot plot showing individual pulmonary histopathology scores for lung samples using the original scoring system (max 24 points). WW vs. SS, Kendall rank order coefficient T = 0.74, z = 8.54, p≤0.00003. NB some similar values that overlapped have been nudged so that the data points can be seen. B. Bland-Altman analysis of the original scoring system WW vs. SS, difference (WW-SS) vs. mean. The dotted lines show the Inter Quarter (IQ) range.

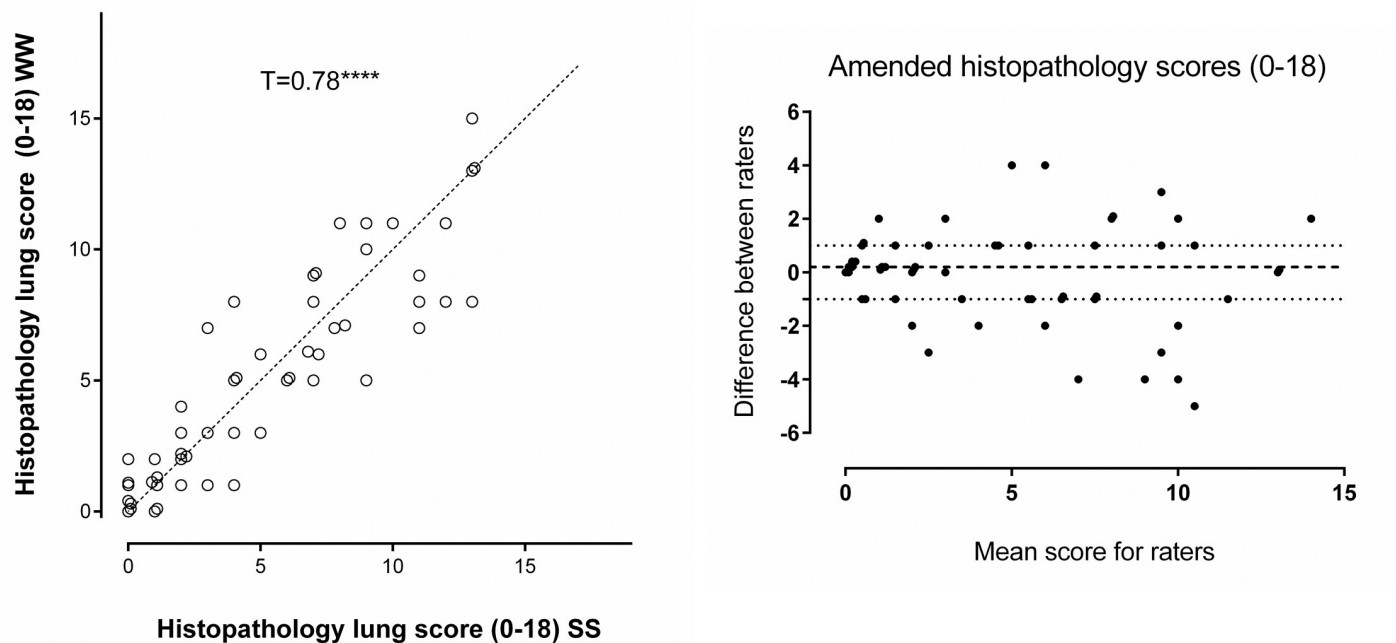

**Fig 3.** A. Scatter dot plot showing individual pulmonary histopathology scores for lung samples using the modified scoring system (max 18 points) WW vs. SS, Kendall rank order coefficient T = 0.78, z = 8.96, p≤0.00003. NB some similar values that overlapped have been nudged so that the data points can be seen. B. Bland-Altman analysis of the modified scoring system, WW vs. SS, difference (WW-SS) vs. mean. The dotted lines show the Inter Quarter (IQ) range.

Overall, the modified scoring system led to good inter- and intra-observer correlations, demonstrating repeatability [32] and is therefore suitable for future minipig OP poisoning studies.

## Discussion

In this work, we have developed a histopathological scoring system specifically for the pulmonary complications of OP insecticide poisoning in minipigs. It is a scoring system that gives repeatable results and is reliable when applied by two scorers. This was possible by combining our current animal model experience with the expertise of two histopathologists with pulmonary and/or pig pathology experience.

The lesions in the lung sections ranged from mild to severe and comprised a broad spectrum compatible with direct and indirect OP insecticide poisoning. The modified histopathology scoring system focusses heavily on the presence of neutrophils in airways, alveoli and pulmonary interstitium. Crucially, it was decided that the histopathologists analysing the slides would calculate the worst possible score per slide, rather than a total score, which meant that the score was independent of the number of airways in each lung section (a feature that is difficult to control between sections). This facilitated more consistent scoring.

Multiple (4–5) lung tissue samples were taken from both caudal and cranial segments of each minipig but they were then treated as individual samples. Although the source of variance of the individual pulmonary histopathology scores would be different [34] than it would had sections originated from separate animals, this consideration is not relevant to the assessment of the ranking, where the source of variance is the observer, not the material.

Aspiration of stomach content causes lung injury in its own right (aspiration pneumonitis), especially when the pH is low (<2.5), the aspirate volume high, or if particulate food matter is

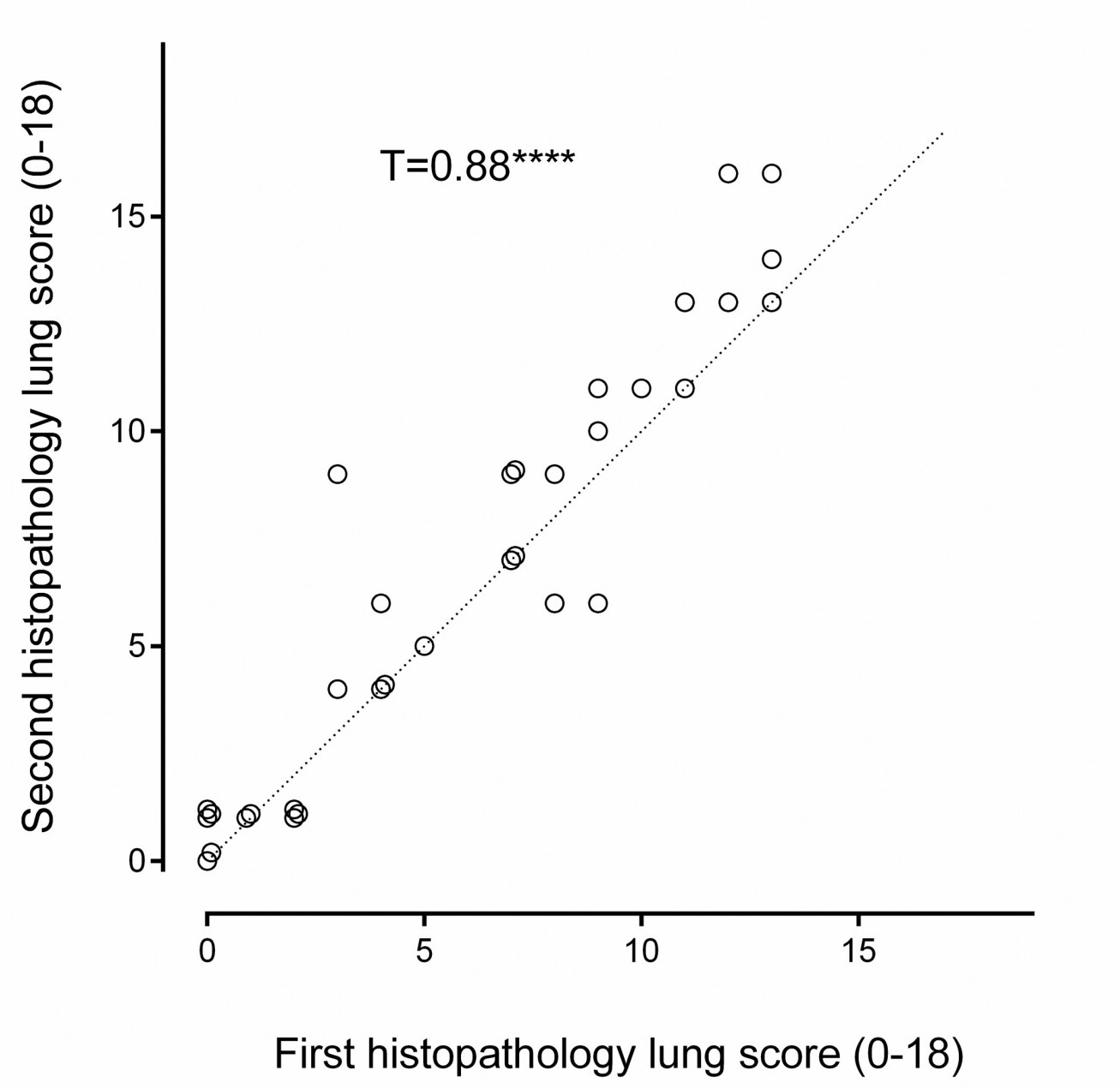

**Fig 4. Scatter dot plot showing individual histopathology scores using the modified scoring system (SS) compared with a second scoring (SS) eight months later (n = 35).** This was to calculate the Intra-observer repeatability of the modified histopathology scoring system. Kendall rank order coefficient T = 0.88, z = 7.44, p≤0.00003. NB some similar values that overlapped have been nudged so that the data points can be seen.

present [35]. Further work in this minipig model will compare pulmonary histopathology after aspiration of gastric content alone versus gastric content plus OP insecticide. Although the scoring system was designed for the study of OP poisoning and gastric content aspiration in minipigs, it might also be applicable to the post-mortem assessment of lung injury in human cases of OP insecticide poisoning, since the anatomy is comparable [18]. In such human studies, links between OP insecticide poisoning, lung injury and death may be possible. Future

work might then focus on which component of the commercial formulation of OP insecticide is responsible for the lung injury and how it might best be avoided (removal from product manufacture) or treated.

## Limitations

The primary limitation of this study is the low number of histopathologists scoring the sections, with only one repeat scoring. Ideally a third histopathologist with no connection to the study should also have scored a subset of sections to determine repeatability. This was mainly due to availability of pulmonary histopathologists with an interest in respiratory and/or porcine pathology. While it would now be possible for us to send scanned images globally to appropriate specialists for their input, this study predates our access to such technology.

Sampling consistency would have been improved by following an approved sampling protocol but this highlights the disadvantage of using pilot study data, as it is often imperfect, designed for honing the main study's protocol for future work. However, in using data from these pilot studies we have maximised the animal model to develop a pulmonary histopathological scoring system that is (a) specific to the model and (b) that will assist our team's future work analysing aspiration and OP induced lung injury.

## Conclusion

In conclusion, we have devised a reproducible scoring system to measure histopathological elements of lung injury caused by orogastric administration of OP insecticides, with or without aspiration of gastric contents.

## Supporting information

**S1 Fig.** A-D. Number of neutrophils in the bronchial lumens. None (score 0 points; Figure A), <10 per airway (score 1 point; Figure B), 11–50 per airway (score 2 points; Figure C), >50 per airway (score 3 points; Figure D).
(TIF)

**S2 Fig.** A-D Number of neutrophils in the bronchiolar lumens. None (score 0 points; Figure A), <10 per airway (score 1 point; Figure B), 11–50 per airway (score 2 points; Figure C), >50 per airway (score 3 points; Figure D).
(TIF)

**S3 Fig.** A-D. Presence of edema in the alveoli/interstitium. None (score 0 points; Figure A), <25% (score 1 point; Figure B), 25–50% (score 2 points; Figure C), >50% (score 3 points; Figure D).
(TIF)

**S4 Fig.** A-D. Numbers of inflammatory cells (mainly neutrophils) in the alveoli. None-few (score 0 points; Figure A), mild increase (score 1 point; Figure B), moderate (score 2 points; Figure C), marked (score 3 points; Figure D).
(TIF)

**S5 Fig.** A-D. Numbers of inflammatory cells (mainly neutrophils) in the interstitium. None-few (score 0 points; Figure A), mild increase (score 1 point; Figure B), moderate (score 2 points; Figure C), marked (score 3 points; Figure D).
(TIF)

**S6 Fig.** A-D. Presence of hemorrhage/necrosis/fibrin anywhere in sample. None (score 0 points; Figure A), up to 5% (score 1point; Figure B), 5–50% (score 2 points; Figure C), >50% (score 3 points; Figure D).
(TIF)

**S1 File. The ARRIVE Essential 10: author checklist.**
(PDF)

**S2 File. Pig lung scores validation Sionagh + William 2019.**
(XLS)

## Acknowledgments

We thank the Wellcome Trust, Mr Adrian Thompson and the veterinary anaesthetic trainees Frances Reed, Rachael Gregson and Ian Self for their support for this study. Professor A. John Simpson is a National Institute for Health Research (NIHR) Senior Investigator. The views expressed in this article are those of the author(s) and not necessarily those of the NIHR, or the Department of Health and Social Care.

## Author Contributions

**Conceptualization:** David A. Dorward, A. John Simpson, Gordon Drummond, Richard E. Clutton, Michael Eddleston.

**Data curation:** Sionagh H. Smith, Michael Eddleston.

**Formal analysis:** Elspeth J. Hulse, Sionagh H. Smith, William A. Wallace, Gordon Drummond, Michael Eddleston.

**Funding acquisition:** Michael Eddleston.

**Investigation:** David A. Dorward, A. John Simpson, Richard E. Clutton, Michael Eddleston.

**Methodology:** Sionagh H. Smith, David A. Dorward, A. John Simpson, Richard E. Clutton, Michael Eddleston.

**Project administration:** Richard E. Clutton, Michael Eddleston.

**Resources:** Michael Eddleston.

**Supervision:** Richard E. Clutton, Michael Eddleston.

**Validation:** Sionagh H. Smith, William A. Wallace, Gordon Drummond.

**Visualization:** Sionagh H. Smith, Gordon Drummond.

**Writing – original draft:** Elspeth J. Hulse, Michael Eddleston.

**Writing – review & editing:** Elspeth J. Hulse, Sionagh H. Smith, William A. Wallace, David A. Dorward, A. John Simpson, Gordon Drummond, Richard E. Clutton, Michael Eddleston.

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
