## [Decision Letter · Decision Letter 0]

7 Sep 2020

PONE-D-20-22461

Development and validation of a histopathology scoring system for the pulmonary complications of organophosphorus insecticide poisoning in a pig model

PLOS ONE

Dear Dr. Hulse,

Thank you for submitting your manuscript to PLOS ONE. After careful consideration, we feel that it has merit but does not fully meet PLOS ONE’s publication criteria as it currently stands. Therefore, we invite you to submit a revised version of the manuscript that addresses the points raised during the review process.

We look forward to receiving your revised manuscript.

Kind regards,

Benito Soto-Blanco, DVM, MSc, PhD

Academic Editor

PLOS ONE

Journal Requirements:

2. As part of your revisions, please thoroughly discuss all methods undertaken to minimize/ameliorate potential pain and distress: environmental enrichment, humane endpoints, monitoring parameters and so forth. Please also complete and submit with your revision the ARRIVE Guidelines checklist - specifically the "Essential 10" checklist: https://arriveguidelines.org/resources/author-checklists.

Reviewers' comments:

Reviewer's Responses to Questions

**Comments to the Author**

1. Is the manuscript technically sound, and do the data support the conclusions?

Reviewer #1: Yes

Reviewer #2: Yes

2. Has the statistical analysis been performed appropriately and rigorously? 

Reviewer #1: Yes

Reviewer #2: Yes

3. Have the authors made all data underlying the findings in their manuscript fully available?

Reviewer #1: Yes

Reviewer #2: Yes

4. Is the manuscript presented in an intelligible fashion and written in standard English?

Reviewer #1: Yes

Reviewer #2: Yes

5. Review Comments to the Author

Reviewer #1: The manuscript presents the results of a pilot (experimental) study with animals (minipigs) to which dimethoate was administered in doses that caused acute intoxication, simulating the ingestion and aspiration of gastric contents. Subsequently, a histopathological scoring system was applied to evaluate the pulmonary complications of the OP-treated minipigs versus controls in order to validate this test. The instrument was valid and replicable to assess the intoxication of pigs by OP poisoning, and it might be used in humans with similar poisonings. In the introduction, the authors link the study with the possibility of improving the treatment of people poisoned by ingestion of OP (suicides).

I believe that this work is relevant. However, I have some observations that should be considered:

- Change the word “organophosphorus” to “organophosphate”

- More background information about the world data on acute pesticide poisonings that are accidental or of occupational origin should be added. Also, an introductory paragraph should be added on the subject of OPs and the worldwide sale of these pesticides. Subsequently, additional literature should be mentioned on the effects at the respiratory level in humans due to exposure to pesticides.

- Add in the method the data analysis plan or procedure used for the validation of the instrument.

- In the introduction, the objective of the study should be clearly stated, starting with an infinitive verb, in order to understand what is the purpose. This would facilitate to evaluate whether the method was well applied.

- In the introduction, I suggest closing with some lines about the relevance of the study, how the results can be applied to understand the effects of acute intoxication in agricultural workers or in situations of self-induced ingestion such as cases of suicide attempt or by accident.

- In the method, before describing the experiments, the type of design must be mentioned and justified. I suggest outlining the procedure of the experiments to facilitate the reader's understanding.

- Sample size is small, justify why

- In the experimental designs: Why did you choose the OP dimethoate? Is it the most used in agriculture? The most toxic? Please justify.

- In the discussion, the results found should be linked with the previous studies cited in the introduction or method.

- The researchers do not state the strengths of the study in the discussion. How it can be applied in humans and the impact of the results on public health is not mentioned. It is not mentioned how methodologically it could be applied in humans.

Reviewer #2: Reviewer’s comments:

PONE D-20-22461:

Development and validation of a histopathology scoring system for the pulmonary

complications of organophosphorus insecticide poisoning in a pig model

General comment:

I would like to congratulate the authors on compiling this contemporary study. As we all know respiratory failure following organophosphorus pesticides (OP) is very common and is the leading cause of death following self-ingestion of such compounds. I found the article interesting and very easy to follow.

However, for it to be accepted and published, I suggest that the following changes should be made to improve the quality of the article.

Comment 1: In line 70 please add a semicolon after NB.

After that I suggest that you add a sentence on the percentages of carbamate, kerosene etc that were found in the solvent the OP compound was dissolved-in to get an idea about the dose response. That way the reader will know that the lung injury was caused by the OP but not the compounds found in the solvent.

Comment 2:

The development of the histopathological scoring system is commendable.

Comment 3:

In the Methods section under the sub sections “ Inter-observer correlation” (line 209) and “Intra-observer correlation” (line 228) , I suggest that you compare the agreement between the different histopathologists and between the same histopathologist by means of Cohen’s kappa statistics and provide a kappa value to quantify the agreement. I believe Krippendorrf’s alpha is a better measurement to compare ordinal values and will give a better agreement. However, even kappa stats can be used for this purpose because it can be done easily in any statistical package that you use. This will give more depth to the message that you all are trying to convey, rather than just stating it is well correlated only. It will solidify the validity of the scoring system in terms of reproducibility.

Comment 4:

Is there any specific reason why you selected only caudal and cranial tissue samples from the lung?

Comment 5:

Discussion is well written. I suggest that you add a few more sentences to elaborate the applicability of this scoring system to human lung tissues based on the a

6. PLOS authors have the option to publish the peer review history of their article (what does this mean?). If published, this will include your full peer review and any attached files.

Reviewer #1: No

Reviewer #2: **Yes: **Dr Chanika Alahakoon (MBBS, MPhil)

---

## [Author Response · Author response to Decision Letter 0]

23 Sep 2020

Dear Editor in Chief,

Thank you for the reviewers’ comments regarding the paper “Development and validation of a histopathology scoring system for the pulmonary complications of organophosphorus insecticide poisoning in a pig model.”

Please find below our answers to the reviewers’ comments (highlighted in yellow). The referenced lines can be found in the ‘marked up’ copy.

Reviewer #1: 

- Change the word “organophosphorus” to “organophosphate”

 ‘Organophosphate’ refers to a specific subgroup of OP insecticides, while the term ‘organophosphorus’ refers to all OP insecticides. ‘Organophosphorus’ is the correct term, and therefore we prefer to keep this term.

- More background information about the world data on acute pesticide poisonings that are accidental or of occupational origin should be added. 

There are currently no papers in print that detail the global incidence of acute accidental/occupational pesticide poisonings. Such a paper is currently being reviewed but has not been published. In the absence of such data, our paper focuses on acute OP pesticide self-poisoning.

Also, an introductory paragraph should be added on the subject of OPs and the worldwide sale of these pesticides.

We have added the following sentences in the introduction to describe the uses of pesticides and an example of financial costings (USA):

“Pesticides have long been used for public health purposes and to protect global food production. The US spends $15 billion a year on pesticides [1], the most common of which is the herbicide glyphosate [2].”

Lines 51-53

-Subsequently, additional literature should be mentioned on the effects at the respiratory level in humans due to exposure to pesticides.

As this paper concerns acute OP poisoning, and not chronic OP exposure, we have referred (in the introduction) to our comprehensive review of respiratory complications in OP poisoning.

“A comprehensive review of respiratory complications secondary to OP compound poisoning is available elsewhere [14].” Lines 75-76

- In the introduction, the objective of the study should be clearly stated, starting with an infinitive verb, in order to understand what is the purpose. This would facilitate to evaluate whether the method was well applied.

We have changed the statement to read:

“The aim of this study was to design a reproducible scoring system to measure both indirect (systemic) and direct (via placement of OP insecticide and gastric juice in lungs) histological lung damage using material from two separate pilot studies conducted by our research team. Lines 90-93

- In the introduction, I suggest closing with some lines about the relevance of the study, how the results can be applied to understand the effects of acute intoxication in agricultural workers or in situations of self-induced ingestion such as cases of suicide attempt or by accident.

We have added the following to the end of the introduction:

“The new pulmonary histopathological scoring system will assist further porcine aspiration studies and could inform observational human OP insecticide poisoning work in the future.” Lines 93-95

- In the method, before describing the experiments, the type of design must be mentioned and justified. I suggest outlining the procedure of the experiments to facilitate the reader's understanding.

We have outlined the whole study at the beginning of the methods section to read:

“The pulmonary histopathology samples for this work were obtained from two large animal pilot studies (Eddleston, in preparation) investigating indirect lung injury through systemic poisoning by gavage (study 1) and simultaneous indirect and direct lung injury through poisoning by gavage and pulmonary placement (study 2) using a common WHO class II moderately toxic OP insecticide, dimethoate.” 

The tissue samples were then scored by two independent histopathologists using a specifically designed scoring system. The reliability of the scoring system was assessed by demonstrating repeatability through inter and intra observer statistical correlation. Lines 98-108.

- Add in the method the data analysis plan or procedure used for the validation of the instrument.

The aim of the work was to design and produce a reproducible scoring system, which we achieved. To make this clearer we have changed the title of the work to remove the word validation and we have changed the following to read:

“The significance of T was calculated through z and if greater than 4 would mean p<0.00003, indicating a very strong relationship between raters’ scores. The inter and intra-scorer correlations indicated the repeatability of the scoring system.[32]” Lines 255-257

- Sample size is small, justify why

Large animal models are expensive to run and this study successfully used clinical information from two pilot studies making best use of previous research animals in accordance with the 3R’s principle – in particular the reduction component. We have added a line to the text:

“Although utilising the clinical information from two previous expensive large animal pilot studies meant accepting low numbers of animals, it is in keeping with the reduction and refinement principle of 3R’s in research.[29]” lines 246-248 

Moreover, although a lack of study power was possible, the result of the test of agreement between the ratings was in both cases very highly significant, indicating that concerns about a false negative conclusion are unfounded.

- In the experimental designs: Why did you choose the OP dimethoate? Is it the most used in agriculture? The most toxic? Please justify.

The Food and Agricultural Organization (FAO) of the UN are shifting use of OPs from Class Ia and Ib (extremely or highly hazardous pesticides) to Class II (moderately hazardous). The most toxic Class II OP is dimethoate (Eddleston M, Eyer P, Worek F, et al. Differences between organophosphorus insecticides in human self-poisoning: a prospective cohort study. Lancet. 2005;366(9495):1452-1459.) 

It is currently used in the UK, some parts of Europe, the US and Australia. https://sitem.herts.ac.uk/aeru/ppdb/en/Reports/244.htm#none

We have changed the intro to the methods to read:

“Pulmonary histopathology samples for this work were obtained from two large animal pilot studies (Eddleston, in preparation) investigating indirect lung injury through systemic poisoning by gavage (study 1) and simultaneous indirect and direct lung injury through poisoning by gavage and pulmonary placement (study 2) using a common WHO class II moderately toxic OP pesticide, dimethoate.” Lines 102-3

- In the discussion, the results found should be linked with the previous studies cited in the introduction or method.

Although we based some aspects of our scoring system on other papers (discussed in intro and methods), we created an entirely new scoring system. Our paper was not about comparing these previous scoring systems with our new one, but to create and validate our own scoring system and so making such comments as suggested may cause confusion.

The actual histopathological findings in study 1 and 2 are being written up now. 

- The researchers do not state the strengths of the study in the discussion. How it can be applied in humans and the impact of the results on public health is not mentioned. It is not mentioned how methodologically it could be applied in humans.

Lines 356-360 explain how the study scoring system might be applied to humans.

 “In such human studies, links between OP insecticide poisoning, lung injury and death may be possible. Future work might then focus on which component of the commercial formulation of OP insecticide is responsible for the lung injury and how it might best be avoided (removal from product manufacture) or treated.” 

Reviewer #2: Reviewer’s comments:

 In line 70 please add a semicolon after NB.

We have done this.

After that I suggest that you add a sentence on the percentages of carbamate, kerosene etc that were found in the solvent the OP compound was dissolved-in to get an idea about the dose response. That way the reader will know that the lung injury was caused by the OP but not the compounds found in the solvent.

This comment appears to refer to lines 73/74. We acknowledge that this information would have been interesting, but the data are not presented in the paper (Kamat SR, Heera S, Potdar PV, Shah SV, Bhambure NM, Mahashur AA. Bombay experience in intensive respiratory care over 6 years. Journal of postgraduate medicine. 1989;35(3):123-134). 

In the Methods section under the sub sections “ Inter-observer correlation” (line 209) and “Intra-observer correlation” (line 228) , I suggest that you compare the agreement between the different histopathologists and between the same histopathologist by means of Cohen’s kappa statistics and provide a kappa value to quantify the agreement. I believe Krippendorff’s alpha is a better measurement to compare ordinal values and will give a better agreement. However, even kappa stats can be used for this purpose because it can be done easily in any statistical package that you use. This will give more depth to the message that you all are trying to convey, rather than just stating it is well correlated only. It will solidify the validity of the scoring system in terms of reproducibility.

We agree that Cohen’s kappa could be used for measuring interobserver variability, but it is generally used more to manage categorical data, whereas Kendall’s test is used typically with ordinal data (Sheskin DJ. Handbook of Parametric and Nonparametric Statistical Procedures, 4th ed, 2007, Chapman and Hall, Boca Raton, p669). Kendall’s test has the advantage that the sampling distribution of the measure (T) approaches normality quickly. This facilitates estimating the probability of observing a T value as extreme as the values we found, if the null hypothesis were true. 

The probabilities we found were gratifyingly small and changed as expected. In conventional P value “money”, values that we found would suggest very strong evidence to reject the null hypothesis (other than possibly in some genetic studies). Presenting our analysis as the probability of the observations is also more familiar and easily understood and interpreted by most readers, rather than Cohen’s kappa. Although of course P values can be calculated for kappa, conventional significance can be reached even when kappa values are disappointingly small. In addition, kappa is affected by the number of and distribution of categories. Finally, interpretation of the kappa value depends on ranges that have been arbitrarily stated: for example 0.61 to 0.80 is classified as “substantial agreement”, or > 0.75 as “excellent”.

The value of Krippendorff’s alpha seems to rest in its generalizability, which wasn’t a substantial requirement in this study. For these reasons we believe that we have used an appropriate test, which we had defined a priori, and would like to retain the original statistical analysis. To clarify the interpretation of the results, we propose that we alter the text at line 254 from:

“The significance of T was calculated through z and if greater than 4, meant that p<0.00003”

to read: “The significance of T was calculated through z and if greater than 4 would mean p<0.00003, indicating a very strong relationship between raters’ scores.” 

Is there any specific reason why you selected only caudal and cranial tissue samples from the lung?

In the pilot studies we were collecting broad evidence of lung injury from the apical and basal lung segments, with the occasional middle lung segment in the right lung. This broadly correlated with sampling each lobe of the lung of the pig to gather as much evidence as possible.

Discussion is well written. I suggest that you add a few more sentences to elaborate the applicability of this scoring system to human lung tissues 

Please see lines 356-360 as per reviewer 1 comments.

Yours faithfully,

Elspeth Hulse PhD FRCA Dip Med Tox

Surgeon Commander Royal Navy

Consultant Anaesthetist, Royal Victoria Infirmary, Newcastle-Upon-Tyne, UK

---

## [Decision Letter · Decision Letter 1]

29 Sep 2020

Development of a histopathology scoring system for the pulmonary complications of organophosphorus insecticide poisoning in a pig model

PONE-D-20-22461R1

Dear Dr. Hulse,

We’re pleased to inform you that your manuscript has been judged scientifically suitable for publication and will be formally accepted for publication once it meets all outstanding technical requirements.

Kind regards,

Benito Soto-Blanco, DVM, MSc, PhD

Academic Editor

PLOS ONE

Reviewers' comments:

Reviewer's Responses to Questions

**Comments to the Author**

1. If the authors have adequately addressed your comments raised in a previous round of review and you feel that this manuscript is now acceptable for publication, you may indicate that here to bypass the “Comments to the Author” section, enter your conflict of interest statement in the “Confidential to Editor” section, and submit your "Accept" recommendation.

Reviewer #1: All comments have been addressed

Reviewer #2: All comments have been addressed

2. Is the manuscript technically sound, and do the data support the conclusions?

Reviewer #1: Yes

Reviewer #2: Partly

3. Has the statistical analysis been performed appropriately and rigorously? 

Reviewer #1: Yes

Reviewer #2: Yes

4. Have the authors made all data underlying the findings in their manuscript fully available?

Reviewer #1: Yes

Reviewer #2: Yes

5. Is the manuscript presented in an intelligible fashion and written in standard English?

Reviewer #1: Yes

Reviewer #2: Yes

6. Review Comments to the Author

Reviewer #1: I thank the authors for the responses. I consider that the manuscript is of quality and they have responded adequately to all comments.

Reviewer #2: The authors have addressed all my comments except my comment on the appropriate statistics. However I still feel there is a place for agreement statistics using kappa or a similar test which is the question I raised in my previous round. This would have improved the quality of the publication But since the authors seems to be in a haste to publish it, the editor can decide about its publication.

7. PLOS authors have the option to publish the peer review history of their article (what does this mean?). If published, this will include your full peer review and any attached files.

Reviewer #1: **Yes: **María Teresa Muñoz-Quezada

Reviewer #2: No

---

## [Editor Report · Acceptance letter]

2 Oct 2020

PONE-D-20-22461R1 

Development of a histopathology scoring system for the pulmonary complications of organophosphorus insecticide poisoning in a pig model 

Dear Dr. Hulse:

I'm pleased to inform you that your manuscript has been deemed suitable for publication in PLOS ONE. Congratulations! Your manuscript is now with our production department. 

Kind regards, 

on behalf of

Dr. Benito Soto-Blanco 

Academic Editor

PLOS ONE